# Cryoablation in Locoregional Management of Complex Unresectable Chest Neoplasms

Roberto Iezzi [1,2], Andrea Contegiacomo [1,*], Alessandro Posa [1], Nico Attempati [1], Ernesto Punzi [1], Alessandro Tanzilli [1], Stefano Margaritora [3,4], Maria Teresa Congedo [3], Alessandra Cassano [5,6], Emilio Bria [5,6], Luca Tagliaferri [7], Vincenzo Valentini [2,7], Cesare Colosimo [1,2] and Riccardo Manfredi [1,2]

1 Dipartimento di Diagnostica per Immagini, Radioterapia Oncologica ed Ematologia, Istituto di Radiologia, Fondazione Policlinico Universitario A. Gemelli IRCCS, L.go A Gemelli 8, 00168 Rome, Italy; roberto.iezzi@policlinicogemelli.it (R.I.); alessandro.posa@gmail.com (A.P.); nicoatt89@gmail.com (N.A.); ernesto.punzi@gmail.com (E.P.); alessandrotanzilli93@gmail.com (A.T.); cesare.colosimo@policlinicogemelli.it (C.C.); riccardo.manfredi@policlinicogemelli.it (R.M.)
2 Istituto di Radiologia, Università Cattolica del Sacro Cuore, 00168 Rome, Italy; vincenzo.valentini@policlinicogemelli.it
3 Dipartimento di Scienze Mediche e Chirurgiche, Unità Operativa Complessa di Chirurgia Toracica, Fondazione Policlinico Universitario A. Gemelli IRCCS, L.go A Gemelli 8, 00168 Rome, Italy; stefano.margaritora@policlinicogemelli.it (S.M.); mariateresa.congedo@policlinicogemelli.it (M.T.C.)
4 Istituto di Patologia Speciale Chirurgica, Università Cattolica del Sacro Cuore, 20123 Milano, Italy
5 Dipartimento di Scienze Mediche e Chirurgiche, Unità Operativa Complessa di Oncologia Medica, Fondazione Policlinico Universitario A. Gemelli IRCCS, L.go A Gemelli 8, 00168 Rome, Italy; alessandra.cassano@policlinicogemelli.it (A.C.); emilio.bria@policlinicogemelli.it (E.B.)
6 Istituto di Medicina Interna e Geriatria, Università Cattolica del Sacro Cuore, 20123 Milano, Italy
7 Dipartimento di Diagnostica per Immagini, Radioterapia Oncologica ed Ematologia, Unità Operativa Complessa di Radioterapia Oncologica, Fondazione Policlinico Universitario A. Gemelli IRCCS, L.go A Gemelli 8, 00168 Rome, Italy; luca.tagliaferri@policlinicogemelli.it
\* Correspondence: andrea.contegiacomo@policlinicogemelli.it; Tel.: +39-3887-924-647

**Abstract:** Rationale and Objectives: The aim of our retrospective study was to assess the safety and feasibility of cryoablation in high-risk patients with complex chest neoplastic lesions. Materials and Methods: Twenty patients with complex chest malignancies, both primary and secondary, located in the mediastinum, lung, and chest wall, underwent percutaneous CT-guided cryoablation treatments. Procedural success as well as complications were evaluated. Results: A total of 24 neoplastic lesions were treated (mean diameter: 27 mm; range: 7–54 mm). Technical success was obtained in all patients, without major complications or intraprocedural death. A pneumothorax not requiring a drainage tube placement was registered in 50% of patients, while 3/24 patients had a grade 3 pneumothorax requiring a chest tube placement. Conclusion: Percutaneous CT-guided cryoablation seems a safe and feasible treatment for complex thoracic lesions.

**Keywords:** cryoablation; lung tumors; interventional radiology

## 1. Introduction

Ablative therapies are an affirmed tool for the management of patients that are not candidates for surgical treatment or with local recurrence after radiation therapy [1]. Most of the literature is focused on Radiofrequency (RFA) and Microwave (MWA) ablation, both characterized by good safety and efficacy [2–4]. However, RFA and MWA suffer some limitations such as the so-called "heat-sink" effect, in lesions close to large vessels, and the inherent risk of thermal damage in lesions proximal to the pleura, the heart, the Aorta, and other mediastinal structures [5]. Cryoablation (CA) induces necrosis using cycles of freezing and thawing and has proven to be a good alternative to RFA and MWA for the management of several cancer subtypes, and grants an overall survival comparable

to RFA and MWA in the lung [3,6]. Cryoablation is also less harmful for normal non-target structures adjacent to the treatment area, and is substantially less painful than RFA and MWA, suggesting that a possible use, in selected patients with severe comorbidities and lesions adjacent to vital structures, is a real technical opportunity, especially in the thoracic district [7].

Based on this background, the primary aim of the present study was to retrospectively evaluate the feasibility and safety of cryoablation in high-risk patients with complex thoracic neoplastic lesions.

## 2. Materials and Methods

This retrospective study was conducted in accordance with the declaration of Helsinki and its amendments; informed consent for this study was waived due to its retrospective nature. All patients signed an informed consent for the procedure at the time of the examination. All consecutive patients with primary and/or metastatic thoracic tumors who underwent cryoablation during the period 2013–2019 were identified by research performed in our institutional radiological information system simultaneously using two keywords ("cryoablation" and "pulmonary tumor"). Only cryoablation procedures performed on patients with complex chest lesions were included in the study; patients with missing procedural data about the applied cryoablation protocol on the radiologic reports or without follow-up data after their discharge from the hospital were excluded.

Pulmonary complex lesions were defined as neoplastic nodules located in unfavorable positions (e.g., in the hilar region of the lungs, strictly adjacent to large mediastinal vessels, cardiac structures, or to larger bronchial branches) and therefore bearing a greater risk of heat-related complications or lower response rates if treated with RFA or MWA.

All the ablative procedures were performed under computed tomography (CT) guidance, referred for cryoablation by a personalized and tailored multidisciplinary tumor board evaluation, including oncologists, thoracic surgeons, radiotherapists, pneumologists, and diagnostic and interventional radiologists.

### 2.1. Procedural Steps

All procedures were performed on an inpatient basis under moderate or deep sedation and strict surgical asepsis. For antibiotic prophylaxis, 2 g cefazoline was administered intravenously before the procedure. Using CT guidance, procedures were performed by two physician authors with 5 and more than 15 years of experience in performing cryoablation procedures, respectively.

Patient's position was assessed by evaluating the previous diagnostic CT examination. A CT scan was acquired to confirm patient position and for needle(s) route planning, once the patient was on the CT scanner bed. After local anesthesia, 14–18-gauge cryoprobes producing unique ice-ball sizes and shapes were used with a commercially available cryoablation system (IceSphere or IceRod probes with the SeedNet cryoablation system; Galil Medical, Yokneam, Israel). The number and type of cryoprobes were selected based on preoperative CT assessment of tumor size, morphology, therapeutic intent, operator's experience, and available materials. When several probes were used, they were spaced 1–2 cm apart to allow a synergistic effect with a fusion of the ice ball generated by each cryoprobe.

A two-cycles freezing-thawing protocol (10 minutes of freezing and 5 minutes of thawing) was always used in all patients. A CT control was performed at the end of each freezing cycle. Needles were removed at the end of the second cycle and a final CT scan was acquired for treatment outcome evaluation and complication check.

Patients' vital parameters (heart rate, respiratory frequency, and blood pressure) were monitored during all the procedures. Conscious sedation was obtained once the needles were within the lesion, and the treatment could be started and was maintained during all the ablation procedures. After the procedure, patients were observed for at least 30 min in a recovery room under constant monitoring. A chest radiography was performed at 3–6 h after the procedure, for identification of late complications; if clinically indicated, or in case

of a positive chest radiography, a subsequent CT scan was performed to better delineate the clinical scenario.

*2.2. Data Collection and Analysis*

The following data were collected: patient characteristics (age, sex, and bleeding risk parameters), tumor characteristics (histology, volume, largest diameter, location, and adjacent anatomical structures), and procedural details (probe type and number, procedural time, technical success, extent of ablation necrosis and perilesional ground glass opacity, complications, and hospitalization length). All variables were evaluated on patients' electronic medical records on the Radiology Information System (RIS) and on the Picture Archiving and Communication System (PACS).

Chart review was performed by an interventional radiologist with 3 years of experience who was blinded to procedural, clinical, and tumor details during data collection.

Lesions were classified as primary if originating from structures within the chest and metastasis if the primary tumor originated in a different district; in addition, lesions were defined as naïve if of new presentation, and recurrence if already previously treated.

Lesion volume was calculated with the ellipsoid formula.

The time between the acquisition of the first CT image after the scanograms and the last control CT scan were used to assess procedural time.

Technical success was defined as the correct needle placement within the tumor and the execution of all the cycles of freezing and thawing as required by the needle vendor. The extent of the ablation necrosis was identified as the low-density, irregular area occurring in the treatment zone close to the end of the procedure, and all three-dimensional measurements of its extent were collected [8]. Concentric or eccentric peripheral zone of ground glass opacity (GGO) occurring after CA of lung lesions was assessed, and its thickness was measured, as it represents the ablation margin and can be a predictive response factor according to Anderson et al. [9].

Hospitalization length was calculated in days, starting from the day of the procedure (included) until the discharge.

The complication rate was reported as the number of complications divided by the number of tumors treated. Complications were classified according to the CIRSE classification of complications and based on Common Terminology Criteria for Adverse Events (CTCAE) version 5.0 [10,11]. Major complications were defined as CTCAE grade 3 or 4, and minor complications as CTCAE grade 1 or 2. Complications were further classified as immediate (<24 h), periprocedural (<30 days), or delayed (≥30 days) [12].

*2.3. Statistical Analysis*

Continuous variables were reported as mean ± standard deviation, and the range of values was also provided in the results paragraph. Categorical variables were provided as absolute numbers and percentages, used for descriptive analysis. Statistical analysis was performed by using SAS version 9.4 (SAS, Cary, NC, USA).

## 3. Results

Among 135 chest ablative procedures performed in our Institution during the study period, 24 procedures performed on 20 patients (mean age: $63.0 \pm 13.2$ years; range: 35–86; 10 males) were eventually included in our retrospective study (Figure 1). Among the 24 treated lesions, 12 (50%) were metastasis and 12 were primary cancers, 15 were naïve lesions, and 9 local recurrence (four post-RT and five post-surgical). Twenty lesions were located in the lung, whereas the last four were located in the mediastinum and chest wall. The mean lesion volume was 8274 mm$^3$ (range: 131–44,012.8 mm$^3$; SD: 11,968.5 mm$^3$) and the mean largest diameter of the lesions was 27.2 mm (range: 7–54 mm; SD: 12.4 mm). Median minimum distance to adjacent critical structures was 1.0 mm (Interquartile range [IQR]: 0.75–2.5 mm). Eighteen out of twenty-four (75.0%) lesions were located in the lung

parenchyma, 3/24 (12.5%) were pleural lesions, and 3/24 (12.5%) were located in the chest wall (Table 1).

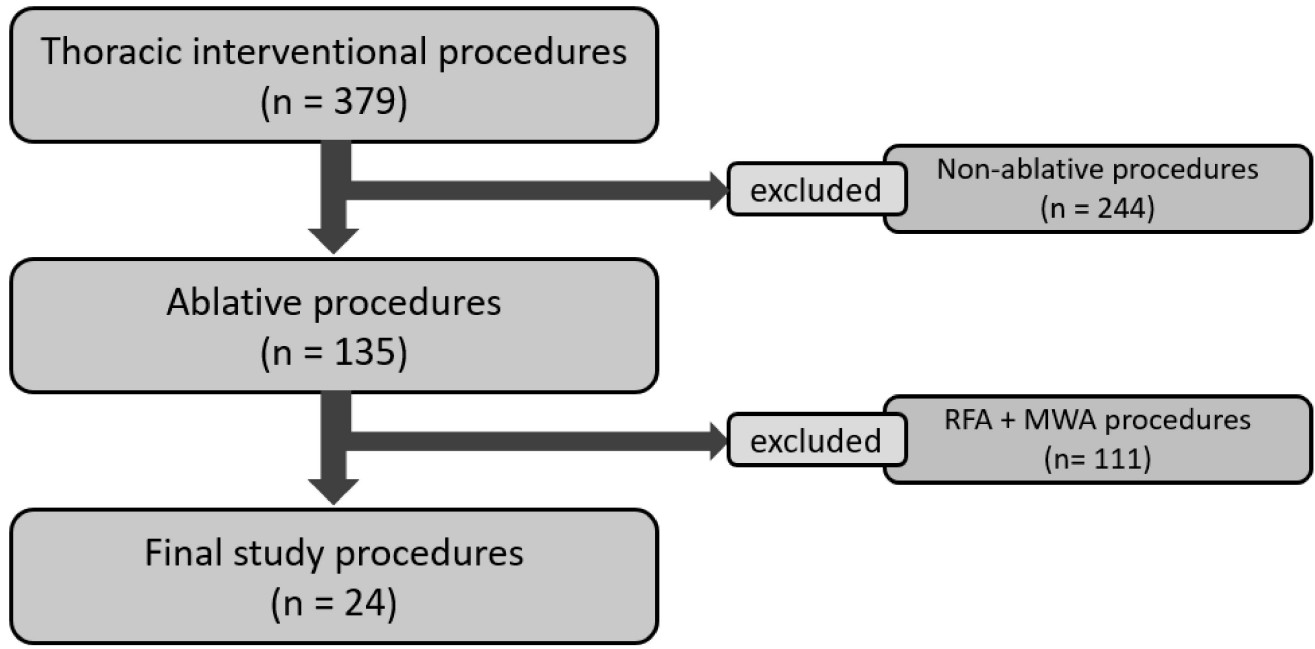

**Figure 1.** Patient selection flow-chart.

Multiple probes were used in 11/24 (45.8%) lesions to obtain adequate coverage of the lesion (median: 1 probe/procedure; IQR: 1–2; range: 1–3). No hydro/gas-displacement or ancillary thermo-protection was usually performed. Technical success was achieved in all the procedures with a mean procedural time of 80 ± 30 min (range 38–144 min).

Mean ablation necrosis diameters, measured on the three axes, and their standard deviations, are reported in Table 2. In 17/20 (85%) cases, a post-treatment perilesional GGO (13/17, 76% concentric; mean thickness 8.8 ± 2 mm) was detected. Median clinical follow-up was 13 months (IQR: 1–29.75 months), with 12/20 (60%) patients with at least 1-year follow-up.

Complications occurred in 13/24 (54.2%) ablative procedures; grade 1–2 pneumothorax was the most frequent (12/24, 50%), followed by grade 1–2 hemoptysis (2/24, 8.3%); no fatal events were observed. In 3/24 (12.5%) procedures, a chest tube placement was necessary to control a grade 3 pneumothorax. No tumor seeding, cryoablation site infection or severe hypotension and arterial bleeding were registered.

Twenty-three out of twenty-four of the registered complications were immediate; one complication was periprocedural. No complications related to the procedure were registered in the follow-up. The median hospitalization length was 3 days (range: 2–10 days). Complications were randomly distributed throughout the study period, with no evidence of a "learning curve" effect, noting that higher-risk cases were undertaken more frequently with increasing institutional experience.

**Table 1.** Study population characteristics.

| Patient Number | Gender | Age | Original Neoplasm | Primary Metastatic | Naïve Recurrency | Lesion Volume (mm$^3$) | Lesion Site | Critical Structure Involved | Minimal Distance (mm) |
|---|---|---|---|---|---|---|---|---|---|
| Patient 1 | F | 68 | Breast | Metastasis | Naive | 360 | Lung | ICV | 11 |
| Patient 2 | F | 60 | Lung | Primary | Recurrence | 4326 | Lung | Aorta | 2 |
| Patient 2 | F | 58 | Lung | Primary | Recurrence | 1608 | Lung | Aorta | 2 |
| Patient 3 | M | 72 | Lung | Primary | Naive | 44,871 | Lung | Pulmonary trunk | 1 |
| Patient 4 | F | 35 | Breast | Metastasis | Naive | 2433 | Chest wall | Heart | 1 |
| Patient 5 | F | 77 | Lung | Primary | Naive | 15,142 | Lung | Thoracic wall | 0 |
| Patient 6 | F | 86 | Lung | Primary | Naive | 5803 | Lung | Thoracic wall | 0 |
| Patient 7 | F | 53 | Melanoma | Metastasis | Naive | 14,601 | Chest wall | ICV | 2 |
| Patient 8 | M | 82 | Lung | Primary | Naive | 6697 | Lung | Pulmonary vein | 8 |
| Patient 9 | M | 66 | Lung | Primary | Recurrence | 1608 | Lung | Azygos vein | 4 |
| Patient 10 | M | 78 | Carcinoid | Primary | Naive | 1485 | Lung | Aorta | 0 |
| Patient 11 | M | 49 | Colon | Metastasis | Naive | 44,012 | Lung | Pulmonary trunk | 1 |
| Patient 12 | F | 63 | Breast | Metastasis | Naive | 5489 | Chest wall | Heart | 0 |
| Patient 13 | F | 46 | Adrenal gland | Metastasis | Naive | 3706 | Lung | Heart | 0 |
| Patient 14 | M | 52 | Liposarcoma | Metastasis | Naive | 131 | Lung | Aorta | 1 |
| Patient 15 | M | 81 | Colon | Metastasis | Naive | 6323 | Lung | Pulmonary vein | 7 |
| Patient 16 | F | 50 | Timoma | Primary | Recurrence | 3161 | Pleura | Left pulmonary vein | 0 |
| Patient 16 | F | 52 | Timoma | Primary | Recurrence | 599 | Lung | Spleen | 7 |
| Patient 16 | F | 52 | Timoma | Primary | Recurrence | 7003 | Pleura | Left kidney | 1 |
| Patient 16 | F | 52 | Timoma | Primary | Recurrence | 15,275 | Pleura | Left kidney | 1 |
| Patient 17 | F | 65 | Lung | Primary | Naive | 3683 | Lung | Aorta | 1 |
| Patient 18 | M | 72 | Rectal | Metastasis | Naive | 5674 | Lung | Heart | 1 |
| Patient 19 | M | 71 | Lung | Primary | Naive | 2995 | Lung | Esophagus | 2 |
| Patient 20 | M | 72 | Lung | Primary | Naive | 1580 | Lung | Pulmonary lobar artery | 8 |

min, minutes; M, male; F, female; ICV, inferior cava vein.

**Table 2.** Procedural characteristics.

| Patient Number | Gender | Age | Needle Number | Needle Brand | Procedural Time (min) | Ice-Ball Extension (mm$^3$) | Ground Glass | Intraprocedural Complications | Clinical Evolution | Hospitalization Length (days) |
|---|---|---|---|---|---|---|---|---|---|---|
| Patient 1 | F | 68 | 1 | IceBall | 63 | 786 | Concentric | - | - | 2 |
| Patient 2 | F | 60 | 3 | IceSphere | 85 | 6864 | Eccentric | Hemoptysis | - | 3 |
| Patient 2 | F | 58 | 2 | IceSphere | 107 | 1996 | Concentric | - | - | 3 |
| Patient 3 | M | 72 | 3 | IceRod | 83 | 39,045 | Concentric | PNX + hemoptysis | Pleural effusion with chest tube placement | 10 |
| Patient 4 | F | 35 | 1 | IceSphere | 80 | 379 | Absent | - | - | 2 |
| Patient 5 | F | 77 | 3 | IceRod | 40 | 14,976 | Concentric | PNX | - | 3 |
| Patient 6 | F | 86 | 2 | IceRod | 74 | 5344 | Concentric | PNX | - | 2 |
| Patient 7 | F | 53 | 2 | IceRod | 93 | 32,531 | Absent | - | - | 2 |
| Patient 8 | M | 82 | 2 | IceRod | 144 | 7657 | Concentric | PNX + pleural effusion | - | 3 |
| Patient 9 | M | 66 | 1 | IceSphere | 75 | 1092 | Concentric | - | - | 2 |
| Patient 10 | M | 78 | 1 | IceRod | 40 | 2009 | Eccentric | PNX | - | 3 |
| Patient 11 | M | 49 | 2 | SeedNet | 127 | 15,670 | Concentric | PNX | Self-limiting hemoptysis | 7 |
| Patient 12 | F | 63 | 1 | IceSphere | 64 | 3557 | Absent | - | - | 2 |
| Patient 13 | F | 46 | 1 | IceRod | 97 | 2177 | Eccentric | - | hemothorax with chest tube placement at 1 day | 7 |
| Patient 14 | M | 52 | 1 | IceSeed | 72 | 772 | Concentric | - | - | 3 |
| Patient 15 | M | 81 | 1 | IceRod | 54 | 6956 | Eccentric | PNX | - | 4 |
| Patient 16 | F | 50 | 1 | SeedNet | 90 | 1383 | Concentric | - | - | 2 |
| Patient 16 | F | 52 | 1 | IceSeed | 38 | 400 | Absent | PNX | - | 3 |
| Patient 16 | F | 52 | 3 | IceSeed (1) + IceRod (2) | 44 | 1521 | Absent | PNX | - | 3 |
| Patient 16 | F | 52 | 2 | IceRod | 45 | 4137 | Absent | - | - | 4 |
| Patient 17 | F | 65 | 1 | IceRod | 60 | 2009 | Concentric | PNX | Chest tube placement | 4 |
| Patient 18 | M | 72 | 2 | IceBall | 63 | 10,025 | Absent | PNX | - | 2 |
| Patient 19 | M | 71 | 1 | IceRod | 45 | 1900 | Concentric | PNX | - | 3 |
| Patient 20 | M | 72 | 1 | IceRod | 63 | 1853 | Concentric | - | - | 2 |

min, minutes; M, male; F, female; PNX, pneumothorax.

## 4. Discussion

Percutaneous cryoablation is a minimally invasive alternative treatment with several properties that make it an attractive ablation option; in detail, it is characterized by a good visualization under CT guidance, preservation of collagenous architecture, and the capability to be performed under local anesthesia [13]. On the other hand, the major limiting factor of cryoablation is the size of the cryoablation zone and the thermal sink effect, which results in a higher local progression rate compared to surgical resection.

In the last years, many published papers highlighted the clinical role of cryoablation, also compared with RFA and/or MWA, but which ablative treatment should be used in a specific patient or a specific lesion is still unclear [3,14–16]. In clinical practice, the decision of the particular ablation technique to use is selected on a case-by-case basis, discussed in a multidisciplinary tumor board [17].

In our institution, it was decided to use cryoablation mainly in the case of unresectable complex chest lesions, usually contraindicated for heating ablation, after discussion in our internal multidisciplinary tumor board: They were tumors located close (<1 cm) to large vessels/hilum/hearth and mediastinal structures in which heat sink effect might lead to thermal ablation failure or to a vessel injury (an example is shown in Figure 2).

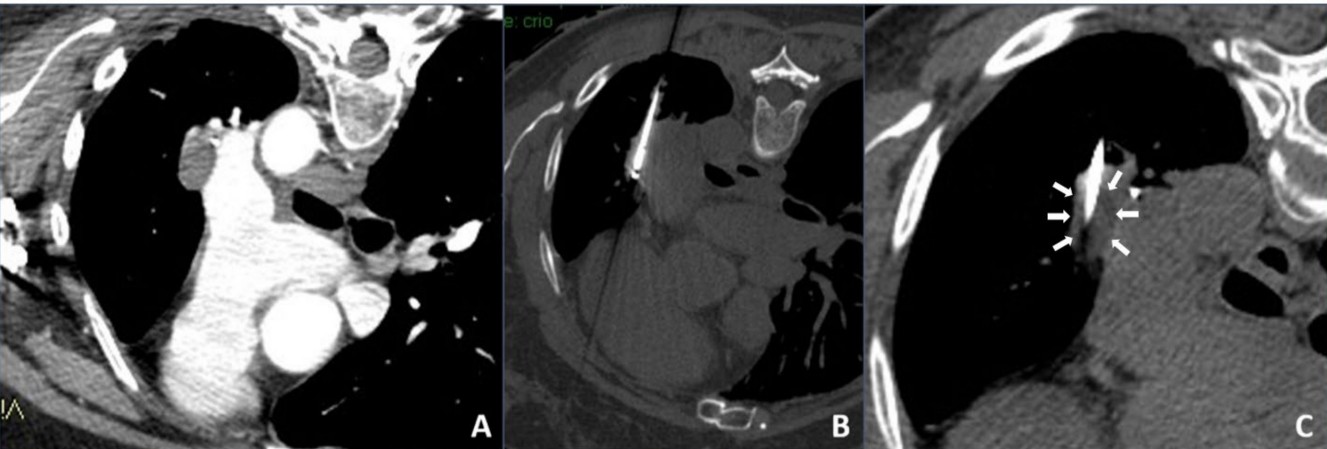

**Figure 2.** Cryoablation procedure: (**A**) contrast-enhanced axial CT scan showing a left hilum neoplastic lesion adjacent to the left pulmonary artery; (**B**) intra-procedural unenhanced axial CT scan with maximum intensity projection (MIP) reconstruction showing the cryoablation needle inside the lesion; (**C**) intra-procedural unenhanced axial CT scan with MIP reconstruction showing direct visualization of the hypodense core necrotic volume.

The lack of a predefined treatment selection in terms of ablative technique justified that, to the best of our knowledge, the safety of cryoablation in unresectable complex chest location had not previously been evaluated and reported. Based on this background, our aim was to retrospectively analyze the safety of cryoablation, in terms of major/minor complication rates, in patients with unresectable complex chest lesions, unsuitable for ablation.

The total complication rate of our study was 54.2%, without fatal events registered. The most common complication was represented by grade 1–2 pneumothorax, registered in 50% of patients, with only 12.5% of procedures in which a chest tube placement was necessary to control a grade 3 pneumothorax.

When considering vascular complications, no severe hypotension and arterial bleeding were registered. In detail, mild pulmonary parenchymal hemorrhage is a universal finding on the immediate CT, usually settling, with significant hemoptysis generally uncommon. In our experience, a grade 1–2 hemoptysis was registered in 8.3% of treated patients, both of them self-limited without any treatment. No tumor seeding or cryoablation site infection were also registered.

The results of our study are in line with previously reported total complication rates, with them mainly being represented by pneumothorax with a reported range of 12–62, even if our lesions were centrally located or closed to hilum, potentially associated with an increased risk of pneumothorax [18]. No pleural effusion was registered also in patients with tumors close to the pleura.

It is mandatory to highlight that all treatments were performed under local anesthesia and/or deep sedation and no significant intraprocedural pain was registered. It is well known that cryoablation is less painful compared with other ablation techniques. Extreme cold acts as an anesthetic, also blocking nerve conduction; furthermore, vasoconstriction of blood vessels from cooling may minimize the resulting edema and reduce the release of pain-inducing substances from damaged tissue.

The main limitation of our study is its retrospective design with a limited number of patients involved. Another limitation could be the short mean follow-up period of the present study; however, it is beyond the scope of the article, which is focused on demonstrating that unresectable complex chest lesions can be treated with locoregional options in a safe way, using cold instead of heat. Despite the statistical limitation of the heterogeneous characteristics of the population and lesion type, including different tumor histology, sizes, and locations, these aspects could be considered as a strength, suggesting the great versatility of cryoablation.

## 5. Conclusions

In conclusion, based on our experience, tumor cryoablation seems to be a safe option for the treatment of complex unresectable chest malignancy with a very low rate of complications. However, our data need to be confirmed by prospective studies obtained in larger populations.

**Author Contributions:** Conceptualization, R.I. and V.V.; methodology, A.C. (Andrea Contegiacomo); software, A.P.; validation, C.C., R.M. and E.B.; formal analysis, A.C. (Andrea Contegiacomo); investigation, N.A. and A.T.; resources, A.C. (Alessandra Cassano); data curation, E.P. and L.T.; writing—original draft preparation, A.P. and A.C. (Andrea Contegiacomo); writing—review and editing, R.I.; visualization, S.M. and M.T.C.; supervision, R.M.; project administration, R.I. All authors have read and agreed to the published version of the manuscript.

**Funding:** This research received no external funding.

**Institutional Review Board Statement:** This retrospective study was conducted in accordance with the declaration of Helsinki.

**Informed Consent Statement:** Informed consent for this study was waived due to its retrospective nature. All patients signed an informed consent for the procedure at the time of the examination.

**Data Availability Statement:** Not applicable.

**Conflicts of Interest:** The authors declare no conflict of interest.

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
