# Peer review of "Cryoablation in Locoregional Management of Complex Unresectable Chest Neoplasms"

_tomography, doi:10.3390/tomography7040057_

Round 1

Reviewer 1 Report

The paper is generally well written and I only have some minor suggestions. 

Line 82-84: It would be better to be a little more specific about the criteria used for determining the number of probes used based on tumor size. I tried to correlate needle numbers with lesion size between Table 1 and 2 but some of them does not look quite correlated. Probably address it more in the writing? 

Line 136 and Figure 1: Where is the number 135 coming from. It is not reflected from Figure 1. Also, for Figure 1, each of the "exclude" should be specify what to exclude right next to it. One thing that really confuses me is whether the "n=##" is the number after the previous exclusion or the number being excluded. "Final Study Population" would better be replaced with "Final Study Procedures". Because these Ns are procedures not patients. 

Any information about the 9 recurrence cases? Are they all local? 

Figure 2C: It would be better to have some arrows showing the necrotic volume region. Also, when it refers to Figure 2 (Line 189) should mention in the text "an example is shown in Figure 2". 

Anything else looks clear to me. 

Author Response

Point 1:  Line 82-84: It would be better to be a little more specific about the criteria used for determining the number of probes used based on tumor size. I tried to correlate needle numbers with lesion size between Table 1 and 2 but some of them does not look quite correlated. Probably address it more in the writing? 

Response 1: The choice of probes number has been based on operator's experience and available material. This information has been added to the manuscript (lines 82-84).

Point 2: Line 136 and Figure 1: Where is the number 135 coming from. It is not reflected from Figure 1. Also, for Figure 1, each of the "exclude" should be specify what to exclude right next to it. One thing that really confuses me is whether the "n=##" is the number after the previous exclusion or the number being excluded. "Final Study Population" would better be replaced with "Final Study Procedures". Because these Ns are procedures not patients. 

Response 2: We have changed the figure 1; I think it's now more readable.

Point 3: Any information about the 9 recurrence cases? Are they all local? 

Response 3: All 9 recurrence cases were local (line 141).

Point 4: Figure 2C: It would be better to have some arrows showing the necrotic volume region. Also, when it refers to Figure 2 (Line 189) should mention in the text "an example is shown in Figure 2". 

Response 4: Figure 2C and line 189-190 have been changed.

Reviewer 2 Report

i do congratulate the authors for this interesting manuscript.

it is a retrospective consecutive series of case (#24 primary or metastatic lung tumors) who received cryoablation as definitive treatment because not-surgical candidates.

design of the study is very clear such as results and discussion

from my point of view there are some critical issues:

  • complication rate is not negligible (>50% of cases with 1 out of 8 cases requiring a chest tube - length of stay was 7days in 2 cases and 10days in one for a procedure that is generally intended on sedation and on out-patient basis). this is a very crucial information coming from this analysis and should be kept in mind when a non surgical indication is seen as a less invasive approach
  • outcomes are procedure-related not influenced by a learning curve (distribution of events is reported as linear) - (I do not have experience but I noted that 45.8% of cases were with multiple probes) - this is another key point that reinforce the general content of this manuscript
  • i do not know the detailed criteria that lead the Authors to favor cryoablation over other treatments (SBRT, RFA, MWA) apart from the risk of thermal injury - I suppose that the concept of "complex location" is a valid explanation though I would suggest to add few lines in the introduction or discussion section to offer a brief review data from the other techniques. 

personally I found this manuscript interesting and in my opinion deserves publication

Author Response

Point 1: complication rate is not negligible (>50% of cases with 1 out of 8 cases requiring a chest tube - length of stay was 7days in 2 cases and 10days in one for a procedure that is generally intended on sedation and on out-patient basis). this is a very crucial information coming from this analysis and should be kept in mind when a non surgical indication is seen as a less invasive approach.

Response 1: I believe this observation is of great importance. In our institution, patients selected for this treatment are NOT eligible for surgical treatment due to multiple comorbidities, so cryoablation is the only option, not an alternative. It is therefore conceivable that the complication rate is higher in our population (even if only 1 procedure required the placement of a thoracic drainage tube), but above all the length of hospitalization which is linked to the "general" fragility of these patients.

Point 2: outcomes are procedure-related not influenced by a learning curve (distribution of events is reported as linear) - (I do not have experience but I noted that 45.8% of cases were with multiple probes) - this is another key point that reinforce the general content of this manuscript

Response 2: The use of multiple probes is linked to the size of the target lesion and to the treatment volume that each probe is able to cover. In larger lesions we used more probes to achieve a larger ablation volume in order to obtain complete response. We think that the use of multiple probes do not requires particular training and that basically the procedure is similar to other ablative techniques so a specific learning curve was not reported.

Point 3: i do not know the detailed criteria that lead the Authors to favor cryoablation over other treatments (SBRT, RFA, MWA) apart from the risk of thermal injury - I suppose that the concept of "complex location" is a valid explanation though I would suggest to add few lines in the introduction or discussion section to offer a brief review data from the other techniques. 

Response 3: a better explanation in the methods section was provided